# Before hands disappear: Effect of early warning visual feedback method for hand tracking failures in virtual reality

**Mucahit Gemici**[1]*, **Vrushank Phadnis**[2], **Anil Ufuk Batmaz**[1]

**1** Department of Computer Science & Software Engineering, Concordia University, Montreal, Quebec, Canada, **2** Google, Mountain View, California, United States of America

* mucahit.gemici@mail.concordia.ca

**Data availability statement:** All data files will be available after acceptance from the "https://osf.io/d5sz6/" database (DOI: 10.17605/OSF.IO/D5SZ6).

## Abstract

Virtual hand representation in Head-Mounted Displays (HMDs) offers immersive and intuitive interactions in Virtual Reality (VR). However, current hand tracking algorithms are prone to errors, which can disrupt the user experience and hinder task performance. This paper presents a novel method for providing users with visual feedback when the quality of hand tracking decreases. Our approach employs a notification modal that warns users of potential failures. We identified three common hand tracking failure scenarios and evaluated the effectiveness of our method in two distinct VR tasks: object manipulation and complex assembly tasks. Results show that our early warning system reduces task completion time, lowers hand-tracking failures by up to 83%, decreases errors, improves system usability, and reduces cognitive load. This work contributes to the development of more robust and user-friendly VR HMD applications by enhancing hand tracking reliability, usability, and workload.

## Introduction

The recent progress in virtual reality (VR) head-mounted displays (HMDs) has made immersive environments more prevalent, allowing users to interact with virtual objects in naturalistic and intuitive ways. One of the biggest challenges in creating such immersive experiences is achieving seamless interaction in the virtual world, especially when using hands, similar to real life. Previous literature has proposed several interaction techniques to provide this experience to the user [1–3]; one of the most well-known and frequently used hand input techniques is called "virtual hand interaction," where the user's real hand and finger movements are directly mapped onto virtual hands and fingers. This direct mapping enables users to easily interact with the virtual world, as they do in real life. Moreover, users rely on accurate and responsive hand tracking to manipulate objects, navigate spaces, and successfully perform tasks when using a VR system with virtual hand interaction techniques.

Current low-cost commercial VR HMD systems track hands using a combination of cameras, sensors, and advanced image processing algorithms [4–6]. Built-in RGB and depth cameras on the VR HMDs or external cameras in the environment capture images of the user's hands. Then, the software uses computer vision algorithms and techniques, often involving

**Funding:** AUB received funding from Google Inc. (https://research.google/programs-and-events/research-scholar-program/). A researcher from the funder contributed to the study design, project administration, validation, and review & editing of the manuscript.

**Competing interests:** The authors have declared that no competing interests exist.

machine learning, to identify and track key points on the hands, such as knuckles and fingertips [7], thus creating a virtual model that mirrors the user's real hand movements in the VR environment.

However, these hand tracking techniques and algorithms are prone to failure. For instance, most machine learning algorithms do not always provide 100% confidence in their predictions [7]. Factors such as fast hand movements [8], occlusions (where one hand blocks the view of the other) [9], or sub-optimal lighting conditions [10] can cause the system to lose track of the hands or make incorrect predictions about their position. When this happens, the virtual hands may not move as expected, which can disrupt the user's experience and reduce the effectiveness of the VR application, as users can not anticipate or quickly adapt to the unexpected behavior of the virtual hands.

When hand tracking algorithms fail, it can disrupt the user's experience and reduce their trust in the system. For example, if the virtual hand does not move as expected, it can lead to task failures or errors. This breaks immersion and creates significant challenges in scenarios requiring precision, such as virtual surgical training or intricate design work. These tracking failures not only hinder task completion but also limit the full potential of VR HMD technology.

Our work investigates the efficiency and effectiveness of a visual feedback method designed to address these issues by warning users *before* hand tracking systems are likely to fail in VR headsets. We monitored hand and finger tracking data of the Meta Quest 3 to detect potential failures. Upon identifying a risk, the system provided users with an early warning, enabling them to prepare and adjust before the error occurred. We evaluated our early-warning method in two user studies that use direct touch interactions, focusing on three critical hand tracking error conditions: low-intensity light, out-of-vision hands, and self-occlusion. The results demonstrate that our approach increases user performance and system usability while reducing workload.

The major contribution of this paper is the design, implementation, and evaluation of a visual early-warning feedback method that proactively alerts users before hand-tracking failures occur in low-cost commercial VR headsets. The primary advancement is shifting from reactive error management toward proactive error prevention in VR hand-tracking interfaces. A flowchart showing the research methodology steps graphically is presented in Fig 1. Specifically, our contributions in this study are the following:

- We propose a VR hand tracking error prediction method for early detection of tracking loss.
- We implement the early warning method as visual feedback to three distinct hand tracking error conditions while detecting potential failures.
- We demonstrate that our method increases user motor performance and improves usability and workload by warning users with a visual modality before hand-tracking failures occur.

## Literature review

### Hand tracking techniques in VR

Hand tracking in VR HMDs has become a crucial component for enhancing user immersion and interaction. To achieve highly accurate hand tracking, several techniques have been proposed, yet the most notable among them is the use of sensor fusion, where multiple sensing devices such as depth cameras, color cameras, and even electromyography (EMG) armbands are combined. Each technology has different limitations, such as camera-based approaches

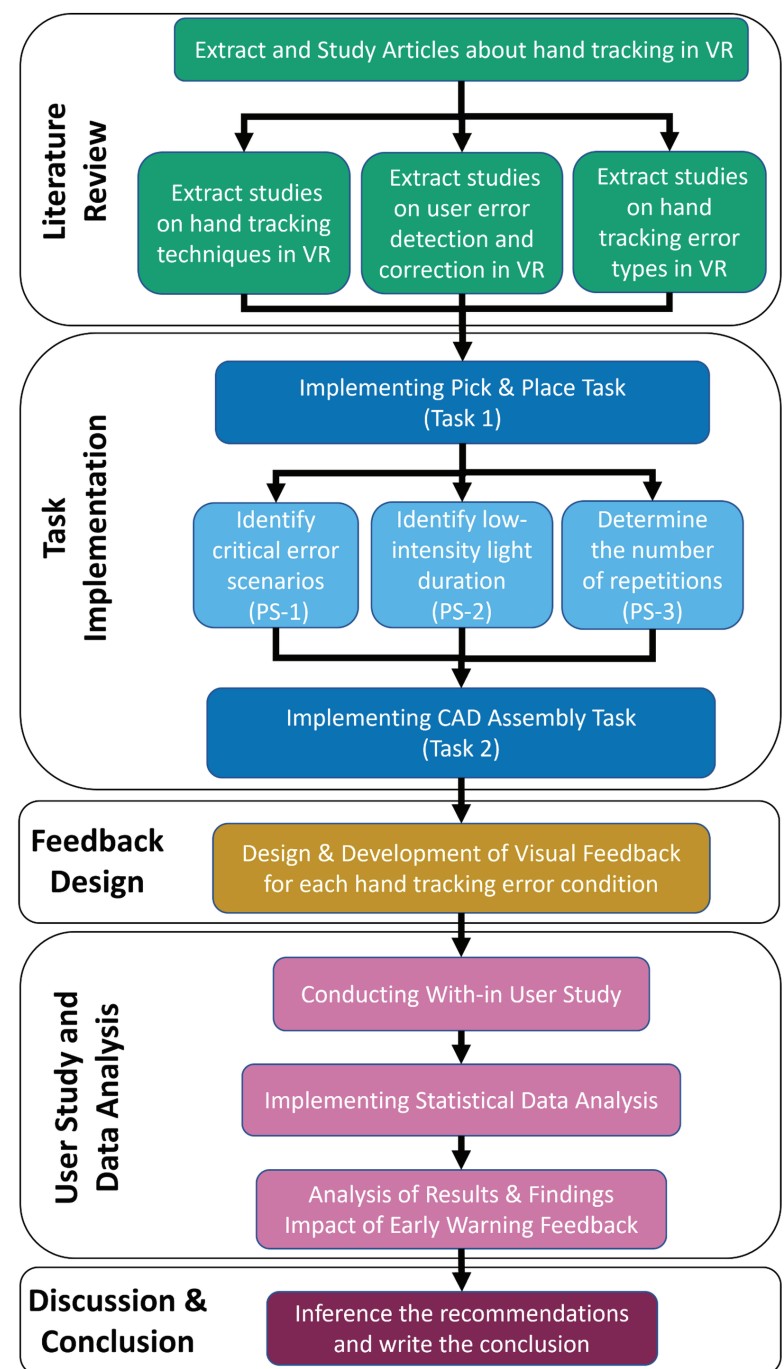

**Fig 1. Research methodology flowchart.** Green represents the literature review, blue task implementation, orange feedback design, pink user study and data analysis, and red Discussion and Conclusion.

are prone to occlusions and lighting sensitivity [11]. The sensor fusion approach helps mitigate common challenges such as occlusions and environmental interference. For instance, Lei et al. [12] proposed a novel sensor fusion system using six Leap Motion controllers, two RealSense depth cameras, and two Myo armbands. This multi-modal setup, combined with

Long Short-Term Memory algorithms, enables precise real-time hand tracking, significantly improving interaction in virtual reality environments.

In addition to sensor fusion, end-to-end differentiable frameworks for multi-view hand tracking have been used for hand tracking. Han et al. [13] developed the UMETrack system, which uses multiple camera views to reconstruct accurate hand poses, effectively addressing occlusions that arise in single-view methods. Similarly, techniques that combine absolute palm tracking with relative finger tracking, as explored by Cameron et al. [14], offer a hybrid approach to improve both the accuracy and usability of hand tracking by fusing different reference frames.

These advances aim to solve the inherent challenges of vision based hand tracking, such as varying lighting conditions, occlusions, and complex hand gestures, ensuring smoother and more immersive experiences for users. Yet, hand tracking in VR remains imperfect and error-prone. To simulate the hand tracking errors, we used Meta Quest 3 headset due to its superior hand-tracking capabilities, which include a wider range of motion and enhanced interaction for complex hand movements, compared to other low-cost headsets currently available on the market [15].

## User error detection and correction in VR systems

Recent research has explored various methods for error detection and correction in VR systems to improve user performance. Gaze dynamics have been shown to effectively distinguish between correctly recognized inputs, input recognition errors, and user errors across different tasks [16]. EEG signals can detect error-related potentials (ErrPs) in response to tracking errors during object manipulation in VR, with an accuracy of 85% [17]. Simulated data analysis has provided insights into error rates and their correlations with user-related factors, such as action duration and fatigue [18]. Additionally, auditory feedback has been found to influence user performance in VR Fitts' tasks, with high-pitch error feedback decreasing performance and adaptive sound feedback reducing error rates without affecting task time [19]. Predicting possible hand movements and detecting erroneous hand actions to show feedback to reduce in-game errors studied by Wolf et al. [20], demonstrating the importance of eye and hand tracking features to provide predictive real-time user support. Moreover, Wang et al. [21] studied a visual feedback approach for hand-object collisions to improve hand movement accuracy. These findings contribute to the development of more effective error detection and correction methods in VR systems. Although error detection and feedback systems have been studied, to the best of our knowledge, prior research has not focused on the effect of feedback visualization before hand tracking system failures. To fill this gap, we explored the effects of an early warning feedback method.

## Hand tracking error types in VR

Previous studies investigated different hand tracking failure conditions in VR to propose novel tracking methods. We categorized these papers under 5 different groups: Occlusion, Environment, System, User, and Other (Fig 2).

**Occlusion** can be broadly categorized in two types, Self-Occlusion [22–31] and External Occlusion [22,23,27,28,32–37]. Self-Occlusion occurs when one part of the hand blocks the view of another part, such as when fingers overlap during a bimanual task. External Occlusion happens when objects outside the hand, like a table or another person, obstruct the camera's view. As an example, Atid and Sutherland [31] discussed how partial or complete occlusion of the hand by the other can occur during bimanual movements. Moreover, Ahmad et al. [28] mentioned that partial and full occlusion as challenges in vision-based hand tracking and

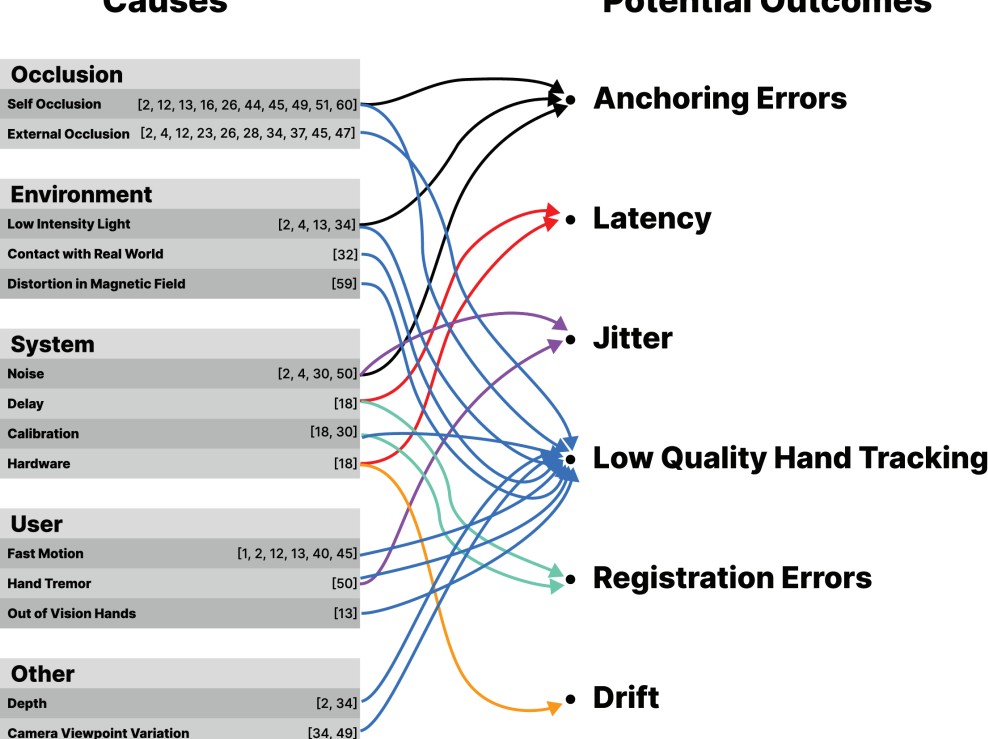

**Fig 2. Cause—potential outcome graph for VR failures**. The causes are grouped under 5 categories: occlusion, environment, system, user, and other. Each category has more than 1 cause and these causes may lead to potential outcomes.

the major obstacle to robust hand tracking in object manipulation scenarios. Lee et al. [22] mentioned both self-occlusion as the occlusion of the fingers, and external occlusion as the occlusion of the hand by the other objects under the topic of 3D pose estimation of hands.

**Environmental** factors pose challenges for hand tracking. Low-light conditions, physical contact with objects [38], and even electromagnetic perturbations can disrupt the tracking process. For example, the presence of a metallic substance can cause a distortion in the magnetic field, affecting the tracking in sensor-based techniques [39]. Erol et al. [23] emphasized the negative effects of uncontrolled environments on hand pose estimation, including varying lighting conditions [28,29,33,36].

**System**-related issues hinder hand tracking accuracy. These include noise [28,36,40,41], delay [42], calibration [41,42], and hardware [42].

**User-induced** errors impact hand tracking accuracy. For example, hand tremor may cause jittery movements [40]. Hands going out of view is a common reason for hand tracking loss [29]. Additionally, fast hand movements, can lead to motion blur, making it difficult for the camera to track accurately [23,27–29,43,44].

The final category of challenges (tagged as **other**) consists of depth and camera viewpoint variation issues. Depth ambiguity is a well-known problem in VR systems, mentioned by Ahmad et al. [28] as a difficulty for vision-based hand tracking and by Mueller et al. [33] as a difficulty for hand pose estimation from RGB videos. Camera viewpoint variation is a vision-based hand tracking problem where the hand is visible from different angles, causing difficulties in hand pose estimation [25,33]. After creating the list of causes, we classified them based on their potential outcomes (Fig 2).

When we categorized the causes of VR hand tracking failures, we also categorized the potential hand tracking failure outcomes based on our literature review (Fig 2). Among the potential hand tracking failure outcomes, e.g., anchoring errors, latency, and drift, we focused on the causes of low-quality hand tracking to develop our feedback method, since it leads the complete hand tracking failures. Low-quality hand tracking was one of the most frequently mentioned in the literature, with its causes well-documented (Fig 2), e.g., occlusions are a major obstacle to hand tracking, particularly in object manipulation scenarios [28]. Additionally, fast hand movements can result in noticeable discrepancies in hand appearance, making accurate tracking difficult [28]. Hand tracking performance also deteriorates as the speed of hand movements increases [45]. Other common causes of low-quality hand tracking include poor lighting conditions [23,28,29,33,36], depth-related issues [28,33], hands moving out of the field of vision [29], and hand tremors [40]. According to Ferstl et al. [29], these hand-tracking failures can result in an erratic motion, reduce the sense of realism, and negatively impact the user experience. We then filtered the low-quality hand tracking failure causes (Fig 2) based on their applicability and importance to low-cost commercial VR-HMDs. For example, we included occlusions as a potential hand tracking error condition due to its frequent investigation in the literature. In Pilot Study 1, we compared the effects of self-occlusion, external occlusion, low-intensity light, real-world contact, and fast motion.

## Motivation

Low-cost VR headsets track hand movements using built-in cameras, but factors like lighting and occlusion can compromise this process. When the hand tracking system fails, it disrupts user tasks, increases errors, and decreases user experience. In everyday computing, notifications (e.g., low-battery alerts) warn users of potential failures before disruptions occur, allowing preventive actions. Similarly, notifying users before hand tracking failures might give them an opportunity to adjust their actions or reposition their hands. This proactive alert can help maintain immersion and reduce frustration. Moreover, when the hand tracking completely fails, users can understand what is happening rather than leaving them unsure why interaction suddenly feels inaccurate or non-responsive.

## Studies

### Task 1 (pick and place task)

Task 1 involved a pick-and-place task as the initial user task to ensure participants could focus on the experience rather than hand-tracking mechanics (Fig 3a). In this gamified toy-world task, participants placed spheres, cubes, square-based pyramids, and plus-signs, as shown in Fig 3c, into target areas. Colors in the task were chosen to be accessible for color-blind users [46–48].

In Task-1, participants were instructed to move objects from a left platform to color-coded areas on a right platform using their dominant hand. Upon the user grabbing one of the 12 objects, a red outline appears around it, a sound is played (Fig 3(b)), and the task begins. When the object is placed correctly in the target area, a success sound is triggered, and the counters for both manipulated and remaining objects are updated (Fig 4). This confirms the successful placement of the object. No feedback was provided for incorrect placements until corrected, after which the success indicators were triggered.

**Pilot studies before Study 1.** Before the main user study, three pilot studies were conducted with a Meta Quest 3 Headset. The first pilot study, conducted from July 15 to July 18, 2024, identified critical hand tracking issues such as low-intensity light, hands out-of-vision,

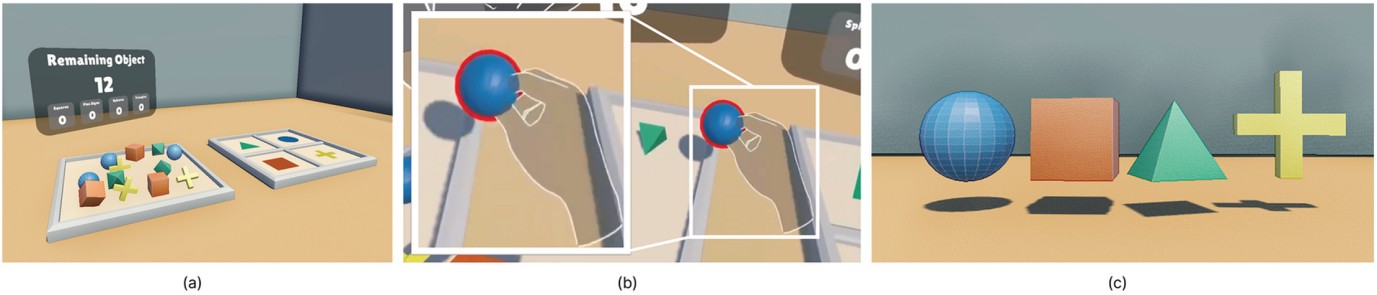

(a)                               (b)                               (c)

**Fig 3. Environment and objects of Task 1.** (**a**) The environment of Task 1. Objects are initialized to the left. Their designated target areas are located at the right and color-coded by category. A counter shows the number of object placements remaining and the error rate for each category. (**b**) A red outline appears around an object when the user grabs it. (**c**) Task 1 objects. From left to right: sphere, cube, square-base pyramid, and plus-sign.

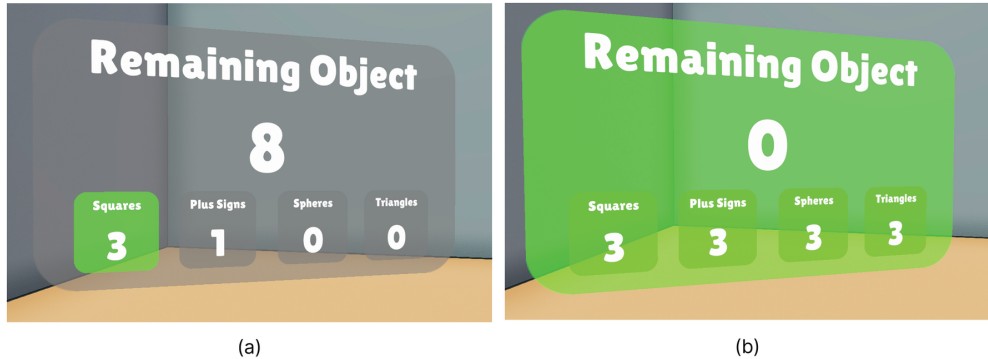

(a)                               (b)

**Fig 4. The user interface (UI) for tracking task progress. The number at the center shows the total number of objects that need to be placed.** Each counter at the bottom is assigned to a specific object group, indicating how many correct placements have been made for that class. (**a**) Once the requirement for a particular group is met, the background of its counter turns green. (**b**) When all objects have been placed in the correct areas, the entire progress UI turns green.

and self-occlusion. The second pilot study, conducted from July 22 to July 24, 2024, determined the best duration for conditions involving low-intensity light. The third pilot study, conducted from August 14 to August 16, 2024, established the number of task repetitions needed.

*Pilot Study 1 (PS-1)* was designed to simulate hand-tracking errors and identify the most critical ones based on our literature review in the related work section (Fig 2). 6 participants completed PS-1 by performing Task 1. To simulate **External Occlusion**, the experimenter placed obstacles between the participant's hand and the VR HMD's cameras, blocking the hand from the cameras. For **Self Occlusion** participants were instructed to hold one hand in front of their face, so that it blocked their view while their other hand interacted with the virtual objects. To simulate **Low-Intensity Light**, the experimenter turned on/off the room's lights as the participants performed the task. For **Out of Vision Hands**, participants were asked to drop objects onto the target from as high as possible. In the **Contact with Real World**, the experimenter periodically placed a water bottle on the desk where the participants were performing the task. To simulate **Fast Motion**, participants were instructed to complete the task as quickly as possible.

Participants in PS-1 completed a questionnaire assessing the impact of different hand tracking error types on their experience, rated from 1 (No Effect) to 7 (Critical Effect), with participants highlighting the most critical error type. The findings indicated that errors **Self Occlusion**, **Low-Intensity Light**, and **Out of Vision Hands** had the highest impact compared to others. Among the six participants, two found **Self Occlusion** most critical, two chose **Out of Vision Hands**, and two selected **Low-Intensity Light**. These errors were also identified as the top three based on task time. Thus, we focused on these three failure types for the remainder of our study.

*Pilot Study 2 (PS-2)* aimed to simulate the Low-Intensity Light error by periodically adjusting room lighting during the experiment. The purpose was to find the minimum duration of low light that does not affect task performance. We compared 2s, 3s, and 4s with 5 participants. Results showed no significant difference in task time or error rate among these durations, suggesting all were effective for simulation. Therefore, we selected a 2s for the Low-Intensity Light condition to minimize disruption while ensuring effectiveness. We chose a 3-second early warning for the Low-Intensity light condition, ensuring participants noticed the feedback before the lights turned off. During PS-2, participants also confirmed they were comfortable with this duration.

*Pilot Study 3 (PS-3)* aimed to determine the number of repetitions for each condition in a user study by focusing on the Out of Vision condition with 4 participants completing 10 repetitions. We found that task time stabilized after 3 repetitions, leading to the decision to use 3 repetitions per condition in the main study. These 3 repetitions were performed *after* training sessions to avoid any potential data instability.

## Task 2 (CAD assembly task)

For Task 2, we selected an assembly task, replicating the mechanical part assembly used by Wang et al. [49]. This task was chosen because it involves direct hand manipulation, aligning with our method's focus on hand-tracking failures. The selected task also provides intuitive and natural interaction with objects, making it a fundamental task in Computer-Aided Design (CAD) [49]. Such tasks are widely used in training systems with proven effectiveness; for example Kalkan et al. [50] report improvements in complex skills by 27.9%, reducing training time by 25%, lower trainer dependency by 82%, and an 89% decrease in assembly defects. This motivated us to choose this task and evaluate our method to demonstrate the effectiveness of the proposed approach in a task that can be applied in real-world practice.

Before starting the task, participants adjusted the platform height to their eye level using the Up, Down, and Confirmation Buttons Fig 5(a). Task 2 required the main body of the part to be aligned with the participants' eye level. Since individuals vary in height, adjusting the platform ensured that everyone had the same perspective, eliminating discrepancies caused by different viewing angles. This adjustment increased accessibility and uniformity across participants. We used a Relative Height that was calculated to indicate the difference between the participants' eye level and the platform height, and this information was displayed on the task state UI as seen in Fig 5(b). Pressing the Up button moved the platform 5 cm upwards while pressing the Down button moved it 5 cm downwards. Participants were instructed to look straight ahead while adjusting the platform height, as any head movement could interfere with the Relative Height calculation. When the Relative Height fell within $\mp$ 2 cm, the background of the Relative Height text turned green, indicating that the height was correctly set, and the Confirmation Button became active. After pressing the Confirmation button, the Relative Height text and buttons are removed and Task 2 began.

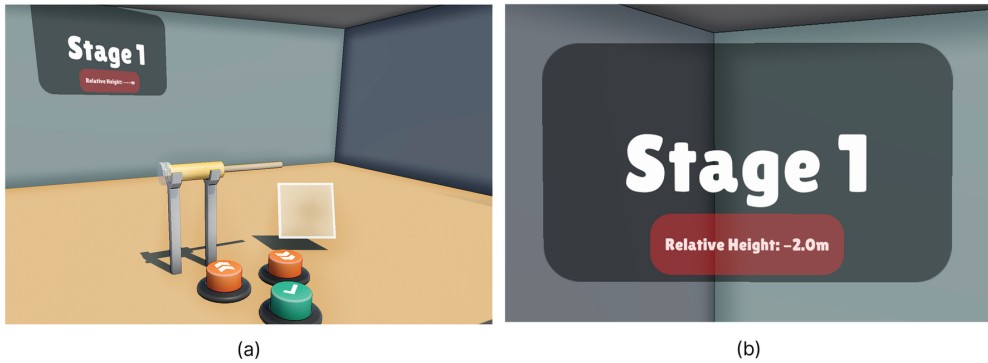

**Fig 5. Starting phase of Task 2.** (**a**) During the height adjustment phase of Task 2, users used the Up and Down buttons in front of them to align the platform to their eye level and pressed the Confirmation Button to finalize the setting. (**b**) The task stage was displayed on a transparent UI, which also guided users in adjusting the platform height to their eye level before starting the task. The UI provided visual feedback, turning the Relative Height background green when the platform matched the user's eye level.

Once Task 2 began, the mechanical parts appeared in the virtual environment, as shown in Fig 6. To ensure accessibility, the colors of the buttons, objects, and environment were chosen to be suitable for color-blind participants.

Once the task began, the first part was highlighted with a thick white outline, as shown in Fig 7a. Participants were instructed to grab the highlighted part and adjust its position and rotation to align with the target area, which was shown with a transparent version of the part (Fig 7b). During the adjustment, the distance and angle between the part and its transparent version were continuously calculated. Alignment was considered successful when the part's distance from the target was less than 4 cm and its rotation angle was within 25 degrees. Once the part met these conditions (Fig 7c), a timer began recording the time the part remained within the specified limits. If the alignment was maintained for 1 second, the part automatically snapped into place in the target area. We simulated the Low Intensity Light Level in the

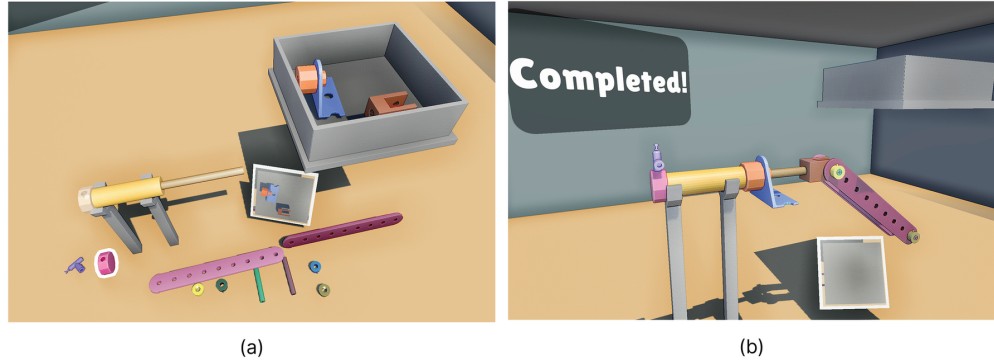

**Fig 6. Top view of the Task 2 environment.** (**a**) Two of the parts were placed in a box located at the top-right area of the user to simulate the Out of Vision error when they attempted to grab these parts. Similar to Task 1, a screen displaying this area was provided so users could monitor their hand position when it was inside the box. (**b**) The completed version of the Task 2.

**Fig 7. Procedure for assembly. (a)** The next part that will be assembled was highlighted with a thick white outline. The outline disappeared when the user grabbed the object. **(b)** The target position for the part was highlighted with the part's transparent visual. The aim was to bring the actual part as close as possible to the transparent version. **(c)** When the both controlled part and it's transparent version were close enough, the transparent version turned into green. After the user was held in this situation for 1s, the part was snapped automatically.

first two stages. In stages 3 and 4, we simulated Out of Vision Hands. Self Occlusion was simulated in stages 8 and 10. We designed the task to simulate each hand tracking error at least two times. After the alignment of each part, the stage number increased. After 10 stages, the task was completed. In stages 8 and 10, 2 screws were assembled at the same time, making the total number of assembly parts 12. After all parts were assembled, the completed version of the mechanical part was displayed Fig 6(b).

## Hand tracking error conditions

In this study, we evaluated the effectiveness and efficiency of an early-warning method across three distinct conditions that we found in PS-1: Low Intensity Light, Out of Vision Hands, and Self Occlusion, and we compared the results with a *benchmark condition*, which we did not simulate any hand tracking error or visual warning.

**Low intensity light condition.** In this condition, the lighting in the room was manipulated while participants performed the task. Initially, the room was set to normal lighting. The experimenter timed lighting events at predetermined intervals, which were not visible to the participant. Six seconds after the task began, the timer signaled "lights off" to the experimenter, who then turned off the lights. After a 2-second interval (as determined in PS-2), the timer signaled "lights on," and the experimenter restored the lighting. This pattern was repeated continuously until the participant completed the task.

**Out of vision hands.** For the Out of Vision Hands condition, we modified the environment by adding virtual walls around the target area platform to enforce indirect manipulation. These walls required participants to move their hands upward to place objects for Task 1 or pick up objects for Task 2. We also placed a virtual screen on the table, displaying a top-down view of the target area (Fig 8a). In this setup, participants had to rely on the 2D screen to accurately position objects in the correct target area. They were instructed to grab objects, move them above the target area (Fig 8b), verify their hand's position on the virtual screen (Fig 8c), and drop the objects from as high as possible while ensuring alignment with the target area. During the task, as participants raised their hands, their hands moved out of the VR-HMD cameras' field of view, inducing the Out of Vision Hands error.

**Self occlusion.** For the Self-Occlusion condition, participants were instructed to hold their non-dominant hand in front of their face, with a visual cue labeled "Hold this hand in front of your face" displayed on the hand (Fig 9). This led to the dominant hand moving

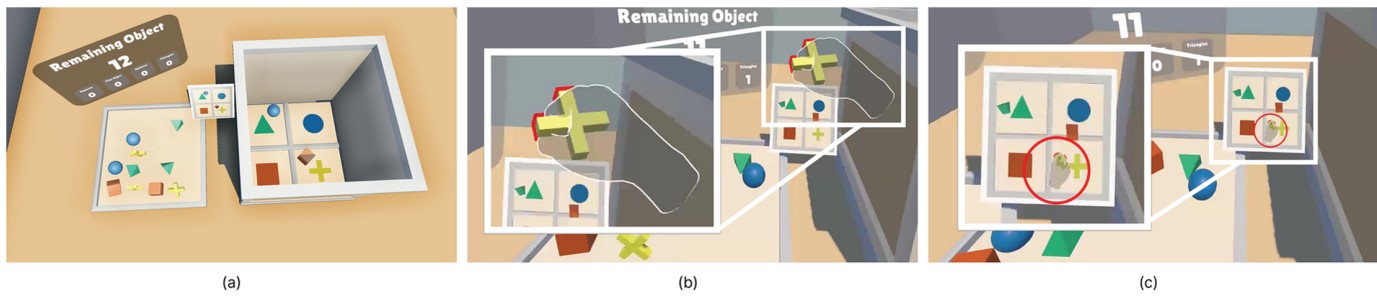

**Fig 8. Task-1 out of vision condition.** (**a**) Participants were instructed to monitor the position of their hand referencing the virtual screen placed in front of them. (**b**) The participant positioned the object above the target area to drop it from the top. (**c**) Throughout the task, participants relied on the virtual screen to verify the hand's position and ensure proper alignment with the target area.

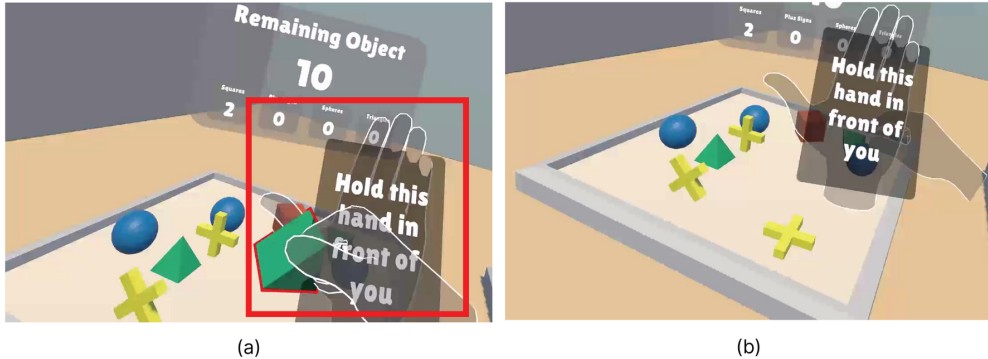

**Fig 9. Self Occlusion hand tracking error scenario.** (**a**) A right-handed participant manipulates the object while holding their left hand in front of their face. This screenshot captures the moment just before a Self-Occlusion hand tracking error occurs. (**b**) When one hand moves behind the other, tracking is lost, causing the object to drop to the ground due to the hand tracking error.

behind the other; causing low tracking quality. The moment before hands are occluded can be seen in Fig 9a. If the occlusion is not fixed, it causes tracking loss, as shown in Fig 9b.

**Modifications of hand tracking error conditions in Task 2.** In Task 1, we individually evaluated each hand tracking error condition, whereas, in Task 2, these conditions were simulated consecutively.

For the Low-intensity light condition, the experimenter changed the lighting conditions on/off in the first two stages. On average, Stage 1 lasted for 19 seconds, and Stage 2 lasted for 11 seconds. We implemented the low-intensity light condition at the beginning of stage 1 and stage 2, so the participants experienced the hand-tracking error at least once. For the Out of Vision condition, two objects were placed inside a box located at the top-right of the participant. Similar to Task 1, a virtual screen was implemented to provide a bird's-eye view of the inside of the box. Participants were instructed to grab the objects from this box by looking at the virtual screen (Fig 10b). To simulate the Self-Occlusion condition, participants were asked to assemble screws simultaneously. The platform was positioned at eye level in front of them, and as they held screws one after another, their hands naturally occluded each other. This resulted in Self-Occlusion errors, as shown in Fig 10a.

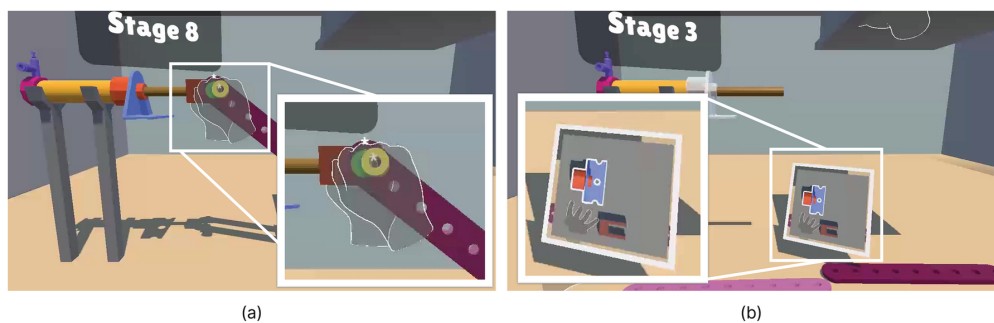

**Fig 10. Self Occlusion and Out of Vision Hands error simulations.** (**a**) Simulating Self Occlusion error. Two screws were instructed to be assembled at the same time and we put a sign on one of the screws to indicate its target area. (**b**) Simulating Out of Vision error. The participants had to look at the screen to grab objects from the box.

## Visual feedback

**Design.** We designed three visual notification modals to warn users of hand tracking errors, consisting of an Error Icon and Text description. A semi-transparent background was used to ensure that users' vision was not obstructed. The Error Icons were specifically designed to represent each error type, and the visual feedback designs are shown in Fig 11. For the Low-Intensity Light error, the feedback included a yellow crossed sun icon and the message "Warning! Low intensity light" (Fig 11(a).) For the Out of Vision error, an orange crossed eye icon was used, with the message "Warning! Hands are not visible" (Fig 11(b).) For the Self-Occlusion error, the feedback featured a red crossed hand icon and the message "Warning! Hands are occluding" (Fig 11(c)).

During the development of the feedback method, we collaborated with an experienced XR designer with 7.5 years of expertise in UI/UX, currently employed at Google. The designer's iteration helped us to improve visualizations. For example, in the initial version of the feedback visuals, all three error icons were red. Following the designer's feedback, we revised the icon colors to enhance differentiation between error types. The updated colors were chosen to align with color-blind accessible palettes, ensuring they were easily distinguishable for all users. There is no weighting between feedback types and icon colors, they are equally important for our feedback method.

**Positioning.** In our study, we explored three methods for positioning warning visuals to ensure visibility: Head Locking, Hand Locking, and World Fixed. Initially, we used Head Locking, placing feedback in the middle-right relative to the participant's head for optimal

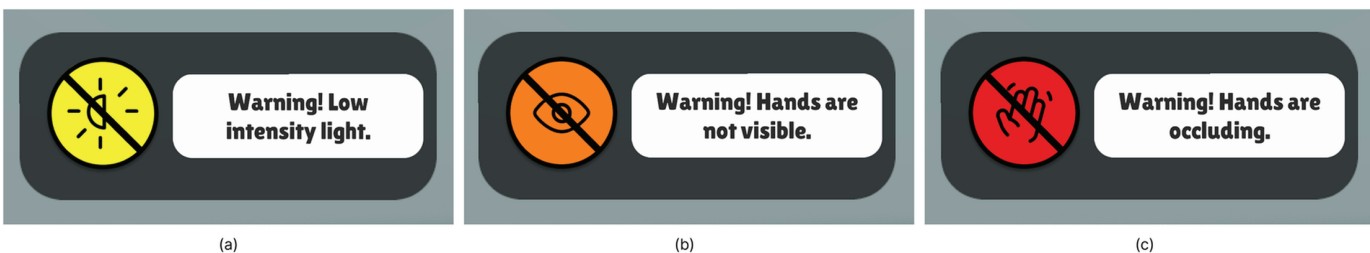

**Fig 11. Visual warning for (a) low intensity light, (b) out of vision hands, and (c) self occlusion.**

visibility [51,52]. However, this led to unnatural scenarios, such as feedback appearing below ground when participants looked down. Afterward, we implemented Hand Locking by attaching the feedback to the participant's dominant hand, but this technique failed during the Out of Vision Hands error, as the feedback became invisible when the hand moved out of view. Finally, we implemented the World Fixed method, placing feedback at a fixed point in the environment (directly in front of the participant) to maintain consistent visibility. After consulting with the XR designer, we chose the World Fixed method. For Task 2 and low intensity light (Fig 12a) and self occlusion (Fig 12c) conditions in Task 1, the feedback was placed directly in front of the user. In the Out of Vision condition in Task 1, the feedback was moved to the top of the virtual screen, as shown in Fig 12b, since users were instructed to look at this screen throughout the task, ensuring the feedback remained visible.

**Error and notification handling.** The aim of the study is to warn users before hand tracking failures occur and evaluate effectiveness of this method. We applied this method to each hand tracking failure condition, before the hand tracking error occurs.

For the Low-Intensity Light condition, the experimenter controlled the external light conditions using a Wizard-of-Oz method to simulate this error. In Unity, we created a counter-based algorithm with Low-Light and Normal-Light states. The state was set to Normal-Light for 6 seconds before switching to Low-Light for 2 seconds. During the last 3 seconds of the Normal-Light state, the software displayed the Low-Intensity Light warning.

To apply our method to Self-Occlusion hand tracking errors, we used the hand position and eye direction. Rays were cast from the user's eye position to each hand, and invisible trigger colliders were placed around the hand models to detect potential collisions. These colliders had two levels: Hand Collider and Hand-Around Colliders. The top view of these colliders is shown in Fig 13a, and the side view is shown in Fig 13b. During the experiment, if one hand moved behind or in front of the other, potentially causing a Self-Occlusion error, the ray intersected the Hand-Around Colliders. This indicated a low risk of Self-Occlusion (Fig 14a). Low-risk situations were ignored to prevent overloading the user with notifications and causing unnecessary distractions. When hands were very close to each other, the ray hit the Hand Collider, indicating a high risk of Self-Occlusion. In this case, we displayed Self-Occlusion feedback, as shown in Fig 14b.

To apply our method to Out of Vision Hands tracking errors, we continuously calculated Forward-Angle, Right-Angle, and RelativePosition-Y. To calculate Forward-Angle, we used the distance vector of the hand and the forward vector of the user's eye. The distance vector of the hand was determined by subtracting the hand's position vector from the user's eye position vector. For Forward-Angle, we used Equation 1, where the hand's distance vector was

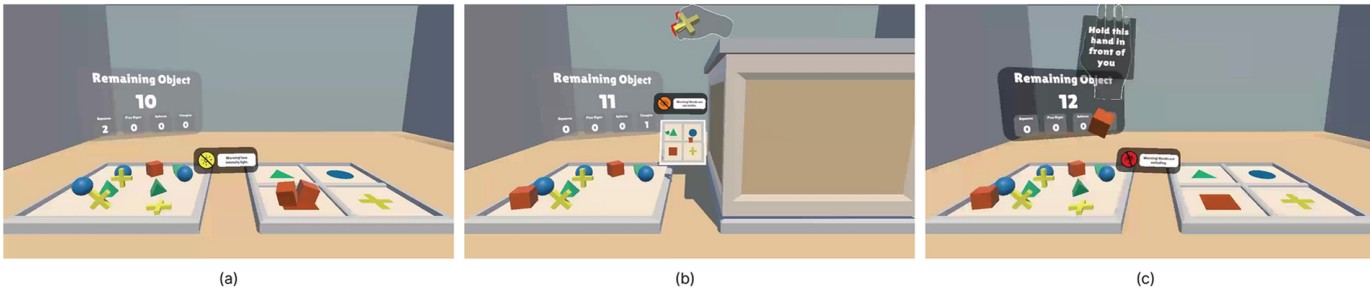

(a)  (b)  (c)

**Fig 12. Visual warning positions for (a) low-intensity light, (b) out of vision hands, and (c) self occlusion conditions.**

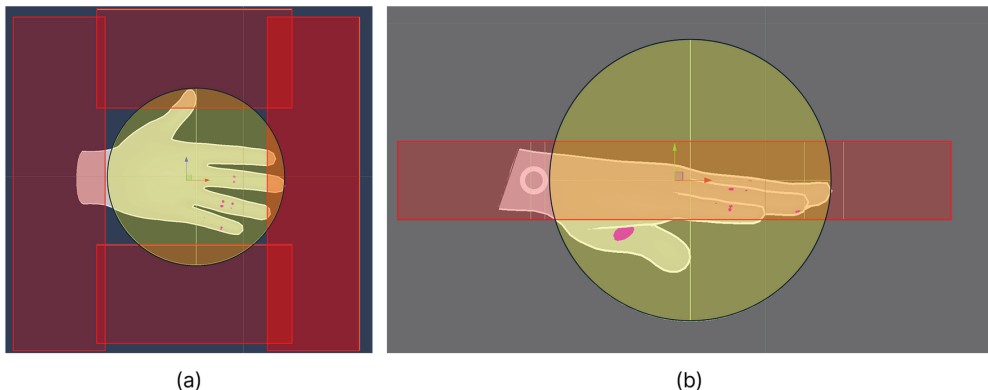

**Fig 13. Visualization of invisible hand colliders from different views.** (**a**) Top-view of invisible hand colliders. The red areas are Hand-Around Colliders and the Yellow areas are Hand Collider. (**b**) Side-view of invisible hand colliders. Red areas are Hand-Around Colliders and the Yellow area is Hand Collider.

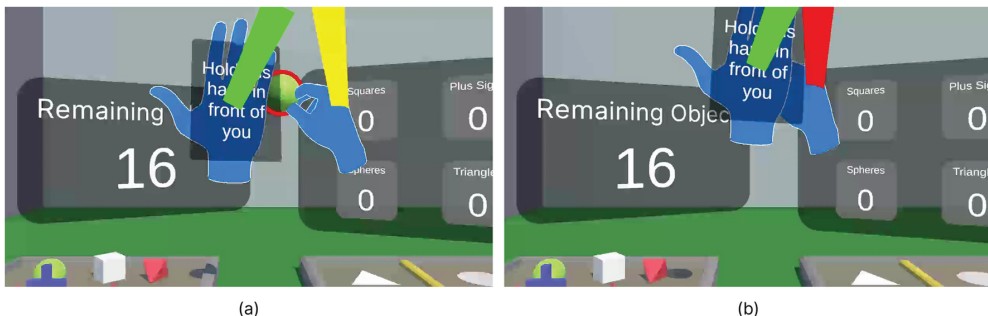

**Fig 14. The lines shown in figures were developed for DEBUG purposes and shown in the figure to explain the method.** The participants did not see any lines during the user study. (**a**) Hands are close to each other. There is a low risk of Self-Occlusion. (**b**) Hands are very close to each other. There is a high risk of Self Occlusion.

represented as $\vec{A}$, and the forward vector of the user's eye was represented as $\vec{B}$.

$$angle = \arccos\left(\frac{\vec{A} \cdot \vec{B}}{\|\vec{A}\| \cdot \|\vec{B}\|}\right) \tag{1}$$

Equation 1 was also used to calculate Right-Angle, with the hand's distance vector as $\vec{A}$ and the right vector of the user's eye as $\vec{B}$. RelativePosition-Y was the y-component of the hand's distance vector. Using the distance vector for RelativePosition-Y helped account for variations in users' arm lengths.

We displayed Out of Vision Hands warning under two conditions. Both conditions were tested before we ran the experiments. The first condition occurred when RelativePosition-Y was greater than or equal to 17 cm, and either Forward-Angle or Right-Angle was greater than or equal to 0.7 degrees. If the first condition was not met, the second condition was triggered when RelativePosition-Y was greater than or equal to 0 cm, and either Forward-Angle or Right-Angle was greater than or equal to 0.6 degrees. In other words, the first condition checks for hand position when they move above the users' head, while the second condition monitors the upper-ipsilateral area.

In Task 1, we evaluated each hand tracking error condition individually, displaying one early-warning feedback at a time. However, in Task 2, if multiple conditions were met simultaneously, all three visual warnings displayed by layering them on top of each other. We also checked the data and ensured that detection accuracy was 100%.

## User study

**Participants.** Our study included 18 participants (9 female, 9 male) aged between 18 and 30 (M = 23.22, SD = 3.08), all of whom were right-handed. Ten participants had corrected-to-normal vision, while the remaining had normal vision. None of the participants were color blind. The details regarding participant demographics, such as their VR interaction experience, are given in the supporting information. This study was approved by Concordia University Human Research Ethics Committee, with certification number of 30019273. The user study was conducted between August 20, 2024, and August 29, 2024. Participants were assigned a written consent form to participate in the user study.

**Apparatus.** The user study was conducted on a desktop PC with an Intel(R) i7-12700F processor at 2.1GHz, 16GB RAM, and an NVIDIA GeForce RTX 3070 graphics card. As the VR Head Mounted Display, we used a Meta Quest 3. The 3D models in the virtual environments were designed using Blender 4.0, and user interface elements were designed using Figma. The VR application was implemented using Unity 2022.3.34f1 with Meta XR All-In-One SDK 66.0.0.

**Experimental design.** We used a within-subjects experiment design for both of our Tasks ($T_2$ = {Task 1, Task 2}). For the Task 1, there were 3 Hand Tracking Error Conditions ($ET_3$ = {Low-Intensity Light, Self Occlusion, Out of Vision Hands}) and 2 Feedback ($F_2$ = {ON, OFF}). In Task 2, participants performed the experiment with Feedback ($F_2$ = {ON, OFF}) and different Hand Tracking Errors based on the task stage. We also compared our results with the **benchmark condition**, where we did not simulate controlled hand-tracking errors or showed any feedback. For both Tasks, we used this uncontrolled —not manipulated— benchmark condition data as the covariant for Repeated Measures (RM) ANCOVA [53]. To avoid order effects, we used counterbalancing with a Latin Square for both tasks.

At the end of each condition, we asked participants to fill out NASA TLX and System Usability Scale (SUS) questionnaires. Friedman and Wilcoxon's tests were used for data analysis of the NASA TLX and SUS questionnaires.

**Procedure.** Upon arrival, participants were briefed about the experiment by the experimenter and given a consent form to read and fill out if they chose to participate. Then, participants were asked to wear the headset and began the training session of Task 1. In the training, participants performed the Task -1 and we did not record data. They tried each hand tracking error condition at least 3 times. They were allowed to repeat the training until they felt comfortable with the task. Following the training, participants performed each hand tracking error condition and feedback condition. Each combination was repeated 3 times. After completing 3 repetitions, participants filled out the NASA TLX and SUS questionnaires. Participants performed this process 6 times (3 hand tracking error conditions × 2 feedback states). Additionally, participants completed the benchmark condition for Task 1 once between blocks of 3 repetitions, and the order of the benchmark condition was counterbalanced across participants.

After Task 1, participants went through the training for Task 2. In training, they first watched the example Task 2 assembly video. Then, they are allowed to perform Task 2 without

recording any data until they feel comfortable. After they confirmed that they were comfortable with the task, we ended the training. Like in Task 1, they also performed a benchmark condition for Task 2, with the order counterbalanced. Participants repeated the Task 2 for 3 times, and then filled out SUS and NASA-TLX questionnaire, as well as a short questionnaire regarding their preferred feedback condition. In summary, the total user study time, including welcoming (5 min), training session for both tasks (3 min + 5 min), benchmark condition (1 min for Task 1, 3 min for Task2), Task 1 (45 min), Task 2 (23 min) was around 85 minutes. The detailed durations are given in the supporting information.

**Evaluation metrics.** Since Meta Quest 3 only provides hand-tracking quality data as a binary variable (LOW-HIGH), it was not enough for us to investigate the effect of the proposed method. Thus, aside from Task Time (TT) and Number of Wrong Placements (N-WP), we collected Number of Low Tracking State for Fingers (NLT-Fingers), Number of Low Tracking State for Hands (NLT-Hands), and Duration of Low Tracking State for Hands (DLT-Hands) data.

**Task Time (TT)** is the total time taken to complete the task. For Task 1, it represents the time from grabbing the first object to placing the last object. For Task 2, TT is the time from grabbing the first mechanical virtual object part to assembling the final object. **Number of Low Tracking State for Fingers (NLT-Fingers)** tracks the sum of low-quality tracking instances for individual fingers per trial, capturing occurrences where the hand tracking system could not track some of the fingers but can predict the untrackable data from the rest of the hand data. **Number of Low Tracking State for Hands (NLT-Hands)** is the record of how many times the hand tracking system failed to track one of the hands per trial. **Duration of Low Tracking State for Hands (DLT-Hands)** is the total duration of the hand tracking lost. While NLT-Hands measures the number of tracking lost, DLT-Hands measures the duration of the tracking lost. **Number of Wrong Placements (N-WP)** measures the number of times participants placed objects into incorrect target areas, i.e., erroneous placement, such as placing a square in the sphere target area. N-WP was used in Task 1.

## Results

The data were analyzed using SPSS 24. We considered it to be normally distributed if Skewness (S) and Kurtosis (K) were within ±1 [53,54]. Otherwise, we performed a log transform to yield a normal distribution. If still not normally distributed, we performed Aligned Rank Transform [55] to analyze data with a parametric test. For post-hoc analyses, we used the Bonferroni method. The figures represent the means and standard error of the mean. After conducting two-way ANCOVA for Task 1, we proceeded to analyze the questionnaire results. Friedmann test was used for NASA TLX and SUS data of Task 1, and then the Wilcoxon test was applied for post-hoc analysis. For Task 2, one-way ANCOVA was conducted, and then the Wilcoxon test was applied for NASA TLX and SUS data. Since we used benchmark condition as an uncontrolled metric independent variable [53], i.e., we did not actively manipulate or systematically regulate as part of study's design, we used RM-ANCOVA for data analysis.

### Task 1 results

In the two-way RM ANCOVA analysis, TT (S = 0.52, K = –0.2), N-WP (S = 0.58, K = –0.58), NLT-Hands (S = 0.01, K = 0.11), and DLT-Hands (S = –0.18, K = –0.6) were normally distributed after log-transform. NLT-Fingers was not normally distributed even after log-transform, so we used Aligned Rank Transform [55]. The results are presented in Table 1 and significant results are shown in Fig 15.

**Table 1. Two-way ANCOVA results for Task 1. The bold values highlight statistically significant results of Task 1.**

|  | Hand Tracking Error | Hand Tracking Error x Benchmark | Feedback | Feedback x Benchmark | Hand Tracking Error x Feedback | Hand Tracking Error x Feedback x Benchmark |
|---|---|---|---|---|---|---|
| **TT** | $F(2,32) = 3.825$ **$p = 0.032$, $\eta^2 = 0.193$** | $F(2,32) = 1.577$ $p = 0.222$, $\eta^2 = 0.09$ | $F(1,16) = 0.005$ $p = 0.946$, $\eta^2 = 0.0$ | $F(1,16) = 0.116$ $p = 0.737$, $\eta^2 = 0.007$ | $F(2,32) = 2.232$ $p = 0.124$, $\eta^2 = 0.122$ | $F(2,32) = 3.202$ $p = 0.054$, $\eta^2 = 0.167$ |
| **NLT-Fingers** | $F(2,32) = 0.920$ $p = 0.409$, $\eta^2 = 0.054$ | $F(2,32) = 1.275$ $p = 0.293$, $\eta^2 = 0.074$ | $F(1,16) = 0.615$ $p = 0.445$, $\eta^2 = 0.037$ | $F(1,16) = 0.341$ $p = 0.568$, $\eta^2 = 0.021$ | **$F(2,32) = 17.660$** **$p <0.001$, $\eta^2 = 0.525$** | $F(2,32) = 2.127$ $p = 0.136$, $\eta^2 = 0.117$ |
| **NLT-Hands** | $F(2,32) = 1.324$ $p = 0.280$, $\eta^2 = 0.076$ | $F(2,32) = 0.267$ $p = 0.767$, $\eta^2 = 0.016$ | **$F(1,16) = 20.445$** **$p <0.001$, $\eta^2 = 0.561$** | $F(1,16) = 0.161$ $p = 0.694$, $\eta^2 = 0.010$ | **$F(2,32) = 15.145$** **$p <0.001$, $\eta^2 = 0.486$** | $F(2,32) = 1.030$ $p = 0.369$, $\eta^2 = 0.060$ |
| **DLT-Hands** | **$F(2,32) = 29.049$** **$p <0.001$, $\eta^2 = 0.645$** | $F(2,32) = 2.939$ $p = 0.067$, $\eta^2 = 0.155$ | **$F(1,16) = 18.504$** **$p <0.001$, $\eta^2 = 0.536$** | $F(1,16) = 0.798$ $p = 0.385$, $\eta^2 = 0.048$ | **$F(2,32) = 15.453$** **$p <0.001$, $\eta^2 = 0.491$** | $F(2,32) = 1.973$ $p = 0.156$, $\eta^2 = 0.110$ |
| **N-WP** | **$F(2,32) = 33.621$** **$p <0.001$, $\eta^2 = 0.678$** | **$F(2,32) = 4.171$** **$p = 0.025$, $\eta^2 = 0.207$** | **$F(1,16) = 27.936$** **$p <0.001$, $\eta^2 = 0.636$** | $F(1,16) = 1.133$ $p = 0.303$, $\eta^2 = 0.066$ | **$F(2,32) = 9.247$** **$p <0.001$, $\eta^2 = 0.366$** | $F(2,32) = 0.308$ $p = 0.737$, $\eta^2 = 0.019$ |

**TT**: According to the results in Table 1, there was a significant main effect of Hand Tracking Error on TT. According to Fig 15, the participants were significantly faster in Low-Intensity Light when compared to Out of Vision Hands and Self Occlusion. Also, they were significantly faster in Self-Occlusion when compared to Out of Vision Hands.

**NLT-Fingers:** Results in Table 1 and Fig 15 showed that NLT-Fingers decreased in out of vision hands and self occlusion conditions when the feedback was visualized.

**NLT-Hands:** The results in Table 1 showed that there was a significant main effect of Feedback on NLT-Hands. According to Fig 15, hand tracking was lost significantly more when there was no feedback.

**DLT-Hands**: The results in Table 1 and Fig 15 showed that there was a significant main effect of Hand Tracking Error on the DLT-Hands. The duration of tracking loss for participants' hands was significantly longer in Low-Intensity Light compared to Self Occlusion. Also, it was longer in Out of Vision Hands compared to Self Occlusion. Moreover, the duration of tracking lost for hands was significantly longer when there was no feedback.

**N-WP:** According to the results in Table 1 and Fig 15, there was a significant main effect of Hand Tracking Error on the N-WP. Participants made fewer mistakes in Low-Intensity Light

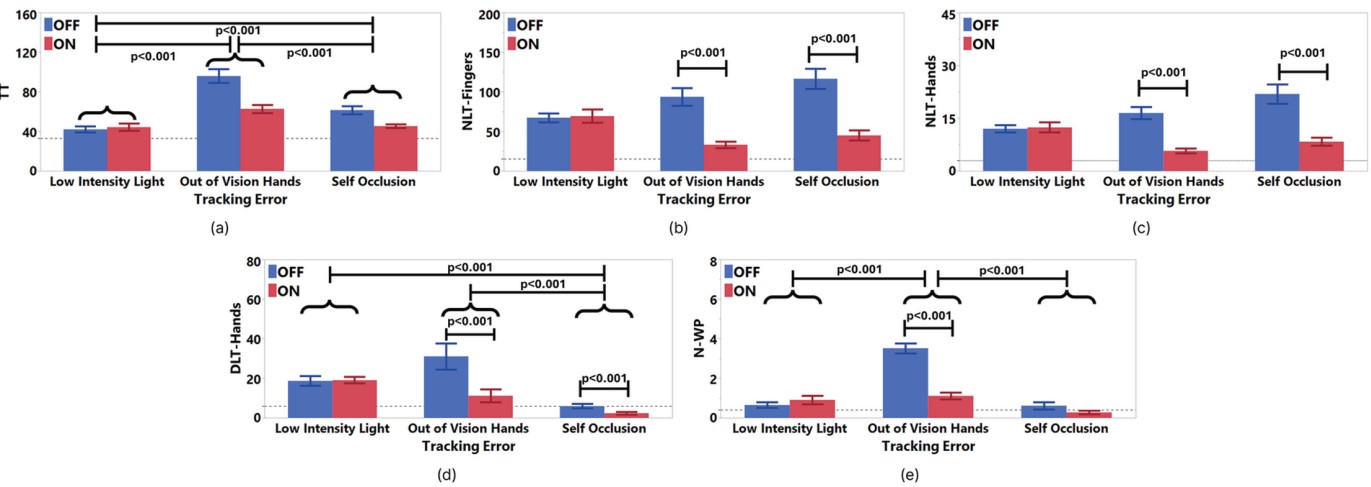

**Fig 15. Task 1 two-way ANCOVA significant results.** Significant results were found for each evaluation metric. The benchmark condition is represented as the dotted line. The metrics shown are (**a**) TT, (**b**) NLT-Fingers , (**c**) NLT-Hands, (**d**) DLT-Hands, (**e**) N-WP.

compared to Out of Vision Hands. Also, they made more mistakes in Out of Vision Hands compared to Self Occlusion. Furthermore, the Feedback had a main effect on the number of wrong placements. Participants placed objects in the wrong areas more when there was no feedback.

## Task 2 results

For Task 2, TT (S = 0.63, K = -0.18), and DLT Hands (S = 0.67, K = 0.07) had a normal distribution after log-transform. NLT-Fingers and NLT-Hands were not normally distributed even after log-transform, so we used Aligned Rank Transform (ART) [55] for their data analysis. The results are presented in Table 2 and Fig 16.

**TT**: According to Table 2, we observed a significant main effect of Feedback on the TT. Participants were significantly slower when there was no feedback (Fig 16(a)).

**NLT-Fingers**: The results in Table 2 showed there was a significant main effect of Feedback on the NLT-Fingers. The NLT-Fingers were significantly higher for Feedback OFF when compared to Feedback ON condition (Fig 16(b)).

**NLT-Hands**: Results in Table 2 showed that there was a significant main effect of Feedback on the NLT-Hands. The hand tracking was lost significantly more for Feedback OFF compared to Feedback ON condition (Fig 16(c)).

**DLT-Hands**: We did not observe any significant difference for DLT-Hands.

**Table 2. One-way ANCOVA results for Task 2. The bold values highlight statistically significant results of Task 2.**

|  | Feedback | Feedback x Benchmark |
|---|---|---|
| **TT** | **F(1,16) = 4.628, p = 0.04, $\eta^2$ = 0.224** | F(1,16) = 3.037, p = 0.101, $\eta^2$ = 0.160 |
| **NLT-Fingers** | **F(1,16) = 33.592, p <0.001, $\eta^2$ = 0.677** | **F(1,16) = 8.411, p <0.01, $\eta^2$ = 0.345** |
| **NLT-Hands** | **F(1,16) = 17.045, p <0.001, $\eta^2$ = 0.516** | F(1,16) = 3.315, p = 0.087, $\eta^2$ = 0.172 |
| **DLT-Hands** | F(1,16) = 0.481, p = 0.498, $\eta^2$ = 0.029 | F(1,16) = 0.094, p = 0.763, $\eta^2$ = 0.006 |

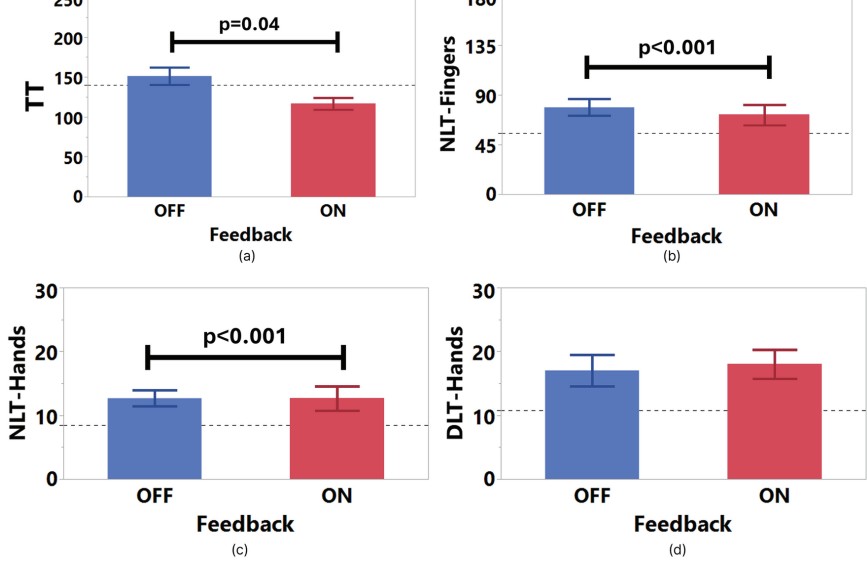

**Fig 16. Task 2 one-way ANCOVA significant results.** Significant differences were found for (**a**) TT, (**b**) NLT-Fingers, and (**c**) NLT-Hands. The benchmark condition is represented as the dotted line. No significant difference was found for (**d**) DLT-Hands.

## NASA TLX results

We found significant differences for Task 1 in terms of NASA TLX scores, such that there were significant differences for Mental Demand ($\chi^2(5, N = 18) = 21.974$, p<0.001), Physical Demand ($\chi^2(5, N = 18) = 29.434$), p<0.001), Performance ($\chi^2(5, N = 18) = 28.767$, p<0.001), Effort ($\chi^2(5, N = 18) = 21.344$, p<0.001), Frustration ($\chi^2(5, N = 18) = 33.489$, p<0.001), and Overall Score ($\chi^2(5, N = 18) = 27.226$, p<0.001).

After the Friedman test, the Wilcoxon test was conducted for post-hoc analysis for the ratings had significant differences. The results are shown in Table 3. The results show that the Feedback ON condition led to significantly more positive ratings compared to the Feedback OFF condition, as shown in Fig 17.

We applied Wilcoxon test to Task 2 NASA TLX data, and we found significant differences for Mental Demand, Physical Demand, Temporal Demand, Performance, Effort, Frustration, and Overall (Table 4). The results presented in Fig 17(d) revealed that all ratings have more negative results for Feedback OFF compared to Feedback ON condition.

**Table 3. Significant Wilcoxon test results of the Task 1 Nasa TLX data, differences between feedback ON and OFF.**

|  | Low Intensity Light | Out of Vision Hands | Self Occlusion |
|---|---|---|---|
| **Mental** | z = −2.147, p = 0.03 | z = −3.159, p <0.01 | z = −2.409, p = 0.01 |
| **Physical** | z = −2.632, p <0.01 | z = −3.35, p <0.001 | z = −2.346, p = 0.01 |
| **Performance** | z = −2.923, p <0.01 | z = −2.587, p <0.01 | z = −2.961, p <0.01 |
| **Effort** | z = −1.993, p = 0.04 | z = −2.875. p <0.01 | z = −2.791, p <0.01 |
| **Frustration** | z = −2.671, p <0.01 | z = −3.419, p <0.001 | z = −3.247, p <0.001 |
| **Overall** | z = −2.548, p = 0.01 | z = −3.68, p <0.001 | z = −3.027, p <0.01 |

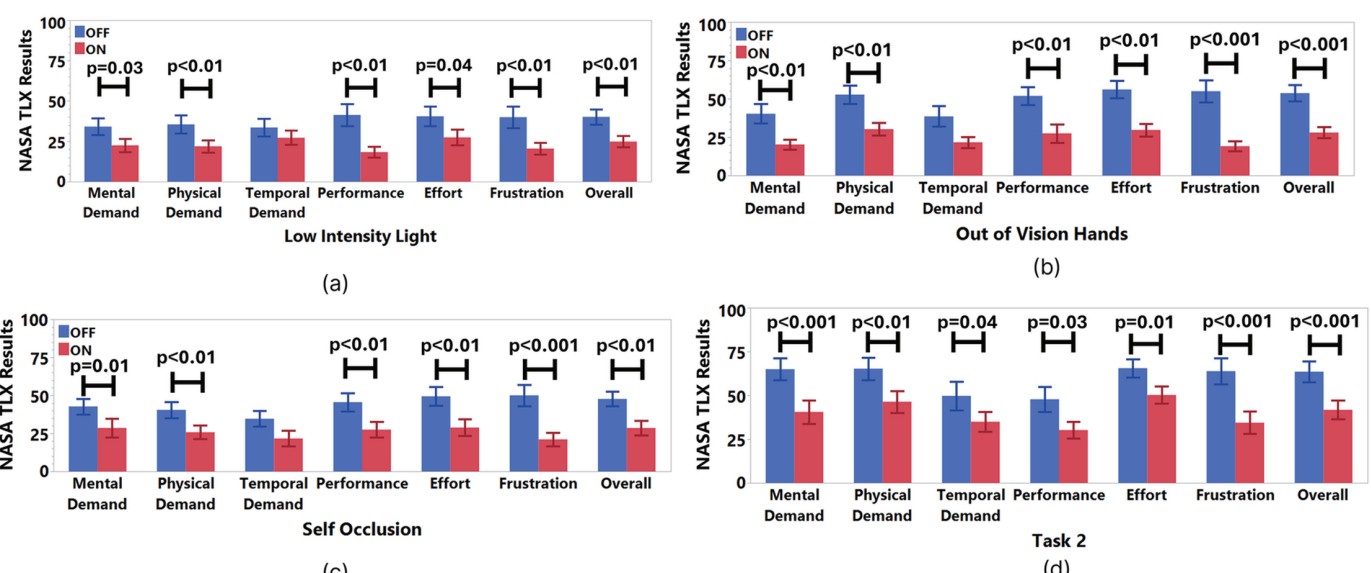

**Fig 17. NASA TLX results of Task 1 for (a) low-intensity light, (b) out of vision hands and (c) self occlusion error.** We found significant differences in all ratings except the Temporal rating. (**d**) NASA TLX Results of Task 2.

**Table 4. Wilcoxon Test results of the Task 2 Nasa TLX data, differences between feedback ON and OFF.**

| Rating | Mental | Physical | Temporal | Performance | Effort | Frustration | Overall |
|---|---|---|---|---|---|---|---|
| Wilcoxon Test Result | z = −3.25, p <0.001 | z = −2.446, p = 0.01 | z =−1.972, p = 0.04 | z = −2.108, p = 0.03 | z = −2.547, p = 0.01 | z = −3.342, p <0.001 | z = −3.246, p <0.001 |

## SUS results

Based on the Task 1 SUS scores shown in Table 5, we observed that Feedback ON conditions got A-Grades, which is defined as an 'Excellent' rating. On the other hand, Feedback OFF conditions got D-Grades, which is defined as a 'Poor' rating. The details for SUS scores, such as analysis for each SUS question, are given in supporting information.

For Task 2, the Feedback OFF condition received a SUS score of 46.25, corresponding to an F-Grade ('Awful' rating). In contrast, the Feedback ON condition achieved a SUS score of 80.42, corresponding to an A-Grade ('Excellent' rating).

## Subjective results

At the end of the experiment, we asked participants which feedback condition they prefer. All 18 participants preferred the early warning method. We also asked for their comments and insights about the user study, and participants commented "It was very helpful in terms of notifying me when a [error] was about to happen," "Feedback was useful," "The feedbacks were [...] placed in a good place where they would be noticed easily yet not hide any of the activity," "The system with feedback was easier to use," "It would help new VR users adapt faster to changes in vision and the environment," "It helps the user navigate the virtual world to achieve a more satisfying experience." Each participant's comments are available in the supporting information.

## Discussion

In this paper, we propose a feedback method that monitors potential VR hand-tracking failures and warns users before hand-tracking failures occur for low-cost HMDs. We applied this method to three critical hand tracking error conditions found by a pilot study: Low-Intensity Light Level, Out of Vision Hands, and Self-Occlusion. We implemented the early warning method for these 3 hand tracking error conditions, and to show its effectiveness, we compared the results with a benchmark condition, where no hand tracking errors are simulated, and feedback off condition, where we simulated the errors but did not display visual feedback. In the early warning method, when a high error risk for one or more hand tracking failures was detected, we warned the users with visual feedback. This enabled participants to recognize the issue, prepare for it mentally, and, if possible, potentially avoid failure by adjusting their behavior.

When the early warning method was enabled, the task time of the participants significantly decreased for the assembly task. We also observed a significant decrease in the number of

**Table 5. SUS score for Task 1.**

| Hand Tracking Error | Feedback OFF (Grade, Score) | Feedback ON (Grade, Score) |
|---|---|---|
| Low Intensity Light | D, 58.47 | A, 87.78 |
| Out of Vision | D, 53.06 | A, 81.25 |
| Self Occlusion | D, 51.38 | A, 83.06 |

hand-tracking and finger-tracking lost in both of the tasks. When the early warning method was activated, tracking of the hands and fingers fared better for Out of Vision Hands and Self Occlusion conditions for both tasks. We also found that the duration of hand-tracking loss was shortened for Out of Vision Hands and Self Occlusion conditions of Task 1. Moreover, we detected that participants made significantly fewer placement errors (placing the object into the wrong area) for the Out of Vision Hands condition of Task 1. These results show that the proposed early-warning method had positive effects on each evaluation metric, depending on the task and hand tracking error condition.

Our analysis of SUS and NASA TLX results further supports the effectiveness of the proposed feedback method compared to the no-feedback condition. For SUS, the usability rating improved from "Poor" to "Excellent" for all hand-tracking error types in Task 1 and from "Awful" to "Excellent" in Task 2, indicating enhanced task usability. NASA TLX results showed reduced mental, physical, and temporal demand, as well as lower performance effort, frustration, and overall workload ratings for the assembly tasks. In Task 1, we observed similar improvements across measures, although temporal demand did not significantly change. Our findings suggest that the early warning feedback method significantly enhances usability and reduces workload across tasks.

We also analyzed the data to see how accurately participants simulated the results when the early warning system provided feedback. Our findings show that the early warning visual feedback was displayed every time before the hand-tracking system failed. The system also showed feedback when it detected a potential hand-tracking failure, even in cases where the system did not actually fail. This pattern is clearly reflected in our data and thus, we observed there was a reduction in the number of hand-tracking errors when early warning feedback was on for out-of-vision hands and self conclusion conditions. We speculate that participants tried to avoid the hand tracking failures when they saw the warning. For instance, one participant commented "I think it is a good feature to know the reasons why the hand motions are not working. The user can adjust their movements depending on what the feedback says, hence can solve the issues." This comment overlaps with our observations during the user study. The experimenter also observed that participants were more careful when the feedback condition was on. Thus, we further analyzed the hand tacking failure results. Based on the average number of tracking lost and feedback, we calculated the feedback coverages for Out of Vision and Self Occlusion conditions, defined as the number of feedback shown divided by the number of hand tracking errors. When the early warning feedback is activated, *hand tracking failures reduced by 83% for the Out of Vision Hands and 62% for the Self Occlusion conditions.* It is important to note that these numbers cannot reach 100%, as the system needed to fail for the task to be completed successfully. However, we speculate that by displaying early warning feedback when a potential failure was detected, *the early warning method helped participants adjust their hand movements more carefully, allowing them to avoid these failures.*

Early warning visual feedback was shown when a hand tracking failure risk was imminent, e.g., in the low-light condition, feedback appeared as the lights dimmed; in the out-of-vision condition, when the hand approached the box; and in the self-occlusion condition, as the hands moved closer together. This allowed participants to mentally prepare or adjust their actions to avoid failure. We speculate that this preparatory phase helped maintain a sense of control, rather than catching them off guard and forcing them to react to an unexpected loss of input fidelity. However, to complete the task, participants had to experience the failure at least once: in low light, the system failed when lights turned off; in the out-of-vision condition, the hands moved out of the headset's tracking zone; and in self-occlusion, one hand blocked the other. We speculate that the feedback helped participants anticipate tracking loss,

mentally prepare for task disruptions, and plan their next steps, making task execution more manageable and reducing frustration. By aligning expectations with reality, the early warning method allowed participants to reframe the failure as something manageable and supported the user's sense of agency rather than a surprise. We think that participants remained engaged, feel more competent, and view the task as easier to handle. This likely decreased cognitive load, as users do not need to rapidly switch strategies on the fly. As a result, their performance improved and system usability increased.

We did not observe any change in the tracking loss for Low-Intensity Light condition even though participants received feedback. This was because when the experimenter dimmed the lights, the hand-tracking system failed completely, leaving participants unable to prevent the failure. In other words, in the Low Intensity Light condition, participants had no control over the environmental lighting, i.e., had no agency over the hand tracking failure, and were unable to prevent hand-tracking errors. Consequently, no significant effects were observed for NLT-Fingers, NLT-Hands, and DLT-Hands. However, the feedback allowed participants to anticipate the failure, mentally preparing themselves for it. This resulted in improved NASA TLX and SUS scores for the Low-Intensity Light condition.

In the Low Intensity Light condition, we adjusted the room's lighting to simulate on/off lights, aiming to maintain consistent effects across trials for analyzing differences between these conditions. However, manual control of lighting changes by the experimenter introduced slight timing inconsistencies due to the manual process. Further, the fully enclosed design of the Meta Quest 3 headset prevented participants from seeing the external lights. Participants were fully engaged in completing the task while managing the simulated hand-tracking errors, which occupied their attention. Consequently, we found no evidence of predictive behavior for Low Intensity Light condition in the data.

We observed that TT was significantly affected by the feedback in Task 2 (Fig 16) but not in Task 1 (Fig 15). We believe this is due to the simplicity of Task 1, which was intentionally designed as a straightforward task to allow participants to focus on the early warning feedback rather than on learning task mechanics. This simplicity may have masked the effect of receiving feedback on TT. In contrast, Task 2, being more challenging, revealed a significant effect of feedback on TT.

Each hand-tracking condition applied in this study represents a real-world scenario. The Low Intensity Light condition simulates sudden lighting changes or power fluctuations, effecting tracking accuracy. Using a dimmer switch, we replicated a scenario where the lights gradually dimmed over 3 seconds before turning off. For instance, in a surgical simulation, if lighting drops unexpectedly due to equipment malfunction [56], early warning feedback can warn the user during the dimming phase, allowing them to be prepared for the tracking failure. To achieve this, we followed the Wizard of Oz method and provided warnings 3 seconds before the lights went out. This approach also applies to scenarios where users experience changes in ambient lighting without being aware of their surroundings, such as when the sun is setting or when moving to a dark room. Similarly, the Out of Vision Hands condition reflects scenarios where users must focus on a point while interacting with objects outside the tracking area, and the Self-Occlusion condition models tasks requiring overlapping hand positions, as in the assembly task.

Additionally, we found that the early warning method decreases the task time for assembly tasks. Previous work developed systems to fix the occlusion problem by adding multiple views [13]. Combining our method with different mechanics can even improve user performance and usability and reduce task workload.

The results of this study can be applied to various fields, such as VR application developers, practitioners, and VR-HMD producers. They can use our feedback method to reduce

user errors caused by hand tracking system failures and provide better usability and lower workload. Our work can be used in studies using direct hand manipulation [57] to increase the usability, and VR medical training [58] where the hand-tracking failures might have a destructive impact. Depending on the application type and interaction techniques it consists of, the method can be modified to include variety of hand tracking error types, specific to the application. For example, the variables to detect Out of Vision hand error condition can be modified for different tasks and systems. In VR surgical planning [59], for instance, different variables may be required due to variations in hand movement areas. For users, our method can also be useful for them such that any failure in an application can break the immersion and affect the user's experience negatively, especially in where high immersion is required [60].

## Limitations and future work

In this study, we applied the feedback method to the three most critical hand-tracking error conditions identified in a pilot study. However, other hand-tracking errors remain unaddressed. We expect future research to investigate additional failure types to broaden the scope of this work.

One limitation of the Meta Quest 3 SDK is its binary hand-tracking confidence level–either LOW or HIGH–rather than a detailed confidence metric. We also tested other devices, including the Meta Quest 2 and Meta Quest Pro, along with various Meta SDKs and Unity's VR solution, but none provided detailed hand tracking confidence data. To overcome this, we developed our custom hand-tracking quality metrics and methods tailored to each error type to predict failures before they occurred. We speculate that integrating built-in device tracking quality variables, if available, could further improve our method's robustness and reliability.

Moreover, we considered using other headsets such as the Meta Quest 2, Meta Quest Pro, and HTC Vive Pro 2. However, since our focus was on low-cost consumer VR HMDs, we selected the Meta Quest 3, which offered the highest inside-out hand tracking performance among available options at the time of the experiment.

We calibrated hand-tracking error failure thresholds for Meta Quest 3. To broaden the reach of our work, different thresholds may be necessary for other headsets. Further, while external devices can measure and mitigate hand-tracking failures, they are often costly and require additional setups, such as external cameras and system calibration, i.e., outside-in tracking. Like the Meta Quest 3, most low-cost consumer VR HMDs rely on integrated cameras for tracking, i.e., inside-out tracking, which we focused on this work.

Our focus was not on resolving hand-tracking errors but on investigating an early warning method to reduce their impact. *We do not propose a definitive solution for hand-tracking errors but instead offer a method to mitigate their effects.*

Simulating hand-tracking errors posed a potential risk of overlapping with internal hand tracking errors, i.e., independent of any deliberate manipulations or simulations. To address this, we used a benchmark condition without simulated errors or feedback in both tasks. As we speculated, we observed hand tracking failures because of internal hand tracking errors. To better isolate and understand the effects of each error condition, we simulated the hand-tracking errors independently in Task 1, comparing the results with benchmark conditions, and found no significant overlap.

In this paper, we focused on hand tracking failures and designed the objects as the participants were required to grab them rather than pinch, such as grabbing a size 1 soccer ball in Task-1. This allowed us to focus on hand tracking failures. Moreover, for instance, participants appreciated that the self-occlusion error simulation was embedded into the task

design (e.g., placing screws) in Task 2 rather than requiring them to hold one hand in front of the other, and this helped them to understand the motivation of this study. We also checked the data and could not find any evidence that such behaviors distracted participants. Yet, we acknowledge that self occlusion errors can also occur when fingers are occluded by the palm, and future studies should investigate this topic.

Our goal was to demonstrate the effectiveness of the early warning method for hand tracking failures, not to provide a universal solution. We acknowledge that the design of the three hand-tracking error conditions was tailored to the tasks, environment, and Quest 3 capabilities. For example, error detection limits for the out-of-vision hands condition were determined through testing. Similarly, we positioned a virtual screen on the table and placed objects out of direct view to induce tracking failures. An HMD with a larger tracking field of view or different tasks may require recalibration of these variables. For other tasks or headsets, these variables should be recalibrated.

In this paper, the gender distribution was equal (9 female, 9 male) and all our participants were right-handed. We recommend that future research investigate user experience and performance across a broader range of demographics. Even though we did not have any color-blind participants, we chose colors that are accessible to potential color-blind participants. We selected the colors based on expert feedback, and we conducted studies at a local university, Concordia University. We recommend future studies take cultural backgrounds into account for selecting colors.

We only explored one form of notification in this work. Previous work found that adaptive sound feedback reduces error rates without affecting task time [19]. Our work uses only visual feedback but confirms that an early warning feedback method positively affects user performance. Thus we recommend future studies to implement different feedback methods, e.g., somatosensory or auditory, to improve user experience and increase user performance.

## Conclusion

This paper presents an early warning feedback method designed to address hand-tracking failures, applied to three specific error conditions across two distinct tasks. Our findings highlight that when participants have control over the source of the hand tracking error, i.e., when they have agency, the feedback system can help them reduce task time, lower hand-tracking failures by up to 83%, make fewer mistakes, experience lower workload, and perceive higher usability. Conversely, when the source of the error is beyond the participants' control, such as environmental conditions like lighting, the feedback still provides benefits by reducing workload and improving usability. We believe that our approach can be valuable for engineers, designers, and VR developers, enabling them to integrate early warning systems for hand-tracking errors into their work. By doing so, they can provide users with a better overall experience when interacting with VR systems.

## Supporting information

**S1 File. Detailed participant demographics.**
(PDF)

**S2 File. ANOVA Results.**
(PDF)

**S3 File. Additional details about the Procedure.**
(PDF)

**S4 File. Detailed SUS results.**
(PDF)

**S5 File. Participant comments.**
(PDF)

## Acknowledgments

We thank the people who participated in our user study, and the UX expert Fyn Ng, for the valuable feedback on our visuals.

## Author contributions

**Conceptualization:** Anil Ufuk Batmaz.

**Data curation:** Mucahit Gemici.

**Formal analysis:** Mucahit Gemici.

**Funding acquisition:** Anil Ufuk Batmaz.

**Investigation:** Mucahit Gemici.

**Methodology:** Mucahit Gemici, Vrushank Phadnis, Anil Ufuk Batmaz.

**Project administration:** Vrushank Phadnis, Anil Ufuk Batmaz.

**Resources:** Anil Ufuk Batmaz.

**Software:** Mucahit Gemici.

**Supervision:** Anil Ufuk Batmaz.

**Validation:** Vrushank Phadnis, Anil Ufuk Batmaz.

**Visualization:** Mucahit Gemici.

**Writing – original draft:** Mucahit Gemici.

**Writing – review & editing:** Mucahit Gemici, Vrushank Phadnis, Anil Ufuk Batmaz.

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
