## [Decision Letter · Decision Letter 0]

PONE-D-25-05813Before hands disappear: Effect of early warning visual feedback method for hand tracking failures in VRPLOS ONE

Dear Dr. Gemici,

Thank you for submitting your manuscript to PLOS ONE. After careful consideration, we feel that it has merit but does not fully meet PLOS ONE’s publication criteria as it currently stands. Therefore, we invite you to submit a revised version of the manuscript that addresses the points raised during the review process.

**ACADEMIC EDITOR:** Please address the issues and revisions of reviewers 1 and 2 before I consider accepting the manuscript.

We look forward to receiving your revised manuscript.

Kind regards,

Ziyu Qi

Academic Editor

PLOS ONE

Journal Requirements:

Additional Editor Comments :

Please address the issues and revisions of reviewers 1 and 2 before I consider accepting the manuscript.

Reviewers' comments:

Reviewer's Responses to Questions

**Comments to the Author**

1. Is the manuscript technically sound, and do the data support the conclusions?

Reviewer #1: Yes

Reviewer #2: Yes

2. Has the statistical analysis been performed appropriately and rigorously? 

Reviewer #1: Yes

Reviewer #2: Yes

3. Have the authors made all data underlying the findings in their manuscript fully available?

Reviewer #1: Yes

Reviewer #2: Yes

4. Is the manuscript presented in an intelligible fashion and written in standard English?

Reviewer #1: Yes

Reviewer #2: Yes

5. Review Comments to the Author

Reviewer #1: The manuscript entitled "Before hands disappear: Effect of early warning visual feedback method for hand tracking failures in VR" represents an advanced technologies failures discussion which is considered noticeable.

The below comments can be noticed for modifications:

1- The major contribution of the research must be explained clearly.

2- As a suggestion, the authors can draw a flowchart to show the research methodology steps graphically.

2- In the title, abbreviations can be used with the complete form of the word, so VR should be written formally (Virtual Reality).

3- On page 2, instead of "Related Work", it is better to use "Literature Review".

4- Figure 15 should be multi-colored to be more representative.

5- In Table 2, F-Score and P-Value must be separated by a comma.

Finally, the manuscript requires at least a minor revision.

Reviewer #2: In the field of VR, hand tracking quality is often compromised by factors such as low-light, occlusion, out-of-view, and fast motion, making it challenging to ensure a 100% tracking accuracy. This manuscript takes a novel approach by proposing preemptive visual feedback to alert users to adjust their interactions before hand tracking failures occur, offering fresh insights for VR product design and development.

Overall, the paper presents complete content with rigorous methodology and experimental design. All critical factors (participant selection, low-light exposure duration, task repetitions, etc.) have been carefully considered. The discussion thoroughly addresses methodological choices and results, with clear awareness of study strengths and limitations. While some limitations exist, they do not undermine the validity of conclusions. The findings provide valuable references for VR product development.

However, certain aspects as following require further clarification and confirmation. After addressing these points, the paper will be suitable for publication.

Considering the above, I recommend minor revision.

1. L262 indicates that low light has been simulated in the first two stages, but L304 only mentions Stage 2. Why is this inconsistent?

2. The task of Stage 2 is to assemble a part. How long does it take? Will the 6s ON : 2s OFF light condition cycle be longer than the task time of this stage, resulting in no occurrence of low illumination in this stage?

3. In the user study procedure, participants repeated the Task 2 for 3 times. But it seems that there’s no comparison between the visual feedback ON/OFF situations. Please explain this point.

4. The benchmark condition does not simulate any hand tracking error or visual warning. So, is tracking failure allowed in actual benchmark experiments? If not allowed, how can we ensure no tracking failures happen in your experiments? In addition, what impact do tracking errors and no tracking errors in benchmark experiments have on the experimental results?

5. Where does the conclusion in L650 come from? From Fig. 14 and 15? But from these two figures, it can be seen that when feedback is turned on, both Task 1 and Task 2 show a significant decrease in TT.

6. In L175, L177, it seems that references to figures should be enclosed in parentheses like in L189, L256, L260, etc.

7. The 'conclusion' in L729 seems to be 'occlusion'.

6. PLOS authors have the option to publish the peer review history of their article (what does this mean?). If published, this will include your full peer review and any attached files.

Reviewer #1: No

Reviewer #2: No

---

## [Author Response · Author response to Decision Letter 1]

21 Mar 2025

Dear Editor,

We would like to thank you and reviewers for your comments. Here is how we addressed each review in the paper:

R1: The major contribution of the research must be explained clearly.

Amendments: In the last paragraph of the introduction, we added a paragraph and clearly stated the major contribution of the work in one sentence:

“The major contribution of this paper is the design, implementation, and evaluation of a visual early-warning feedback method that proactively alerts users before hand-tracking failures occur in low-cost commercial VR headsets. The primary advancement is shifting from reactive error management toward proactive error prevention in VR hand-tracking interfaces.”

R1: As a suggestion, the authors can draw a flowchart to show the research methodology steps graphically.

Amendments: We added a flowchart showing the research methodology at the end of the introduction section.

R1: In the title, abbreviations can be used with the complete form of the word, so VR should be written formally (Virtual Reality).

Amendments: We fixed the title as the reviewer suggested.

R1: On page 2, instead of "Related Work", it is better to use "Literature Review".

Amendments: As the reviewer suggested, we changed the title of the subsection to “Literature Review.”

R1: Figure 15 should be multi-colored to be more representative.

Amendments: We changed the color of Figure 16 (Figure number is now changed to 16) to multi-color, as the reviewer suggested.

R1: In Table 2, F-Score and P-Value must be separated by a comma.

Amendments: We added a comma between F-scores and P-Values in Table 2.

R2: L262 indicates that low light has been simulated in the first two stages, but L304 only mentions Stage 2. Why is this inconsistent?

Amendments: We thank the reviewer for their comment. The participants experienced the low-intensity light condition in the first two stages of Study 2. We fixed the text on L304.

R2: The task of Stage 2 is to assemble a part. How long does it take? Will the 6s ON : 2s OFF light condition cycle be longer than the task time of this stage, resulting in no occurrence of low illumination in this stage?

Amendments: On average, stage 1 lasted for 19 seconds, and stage 2 lasted for 11 seconds. We implemented the low-intensity light condition at the beginning of the task (stages 1 and 2), so the participants experienced the hand-tracking error at least once. This is now mentioned in the text.

R2: In the user study procedure, participants repeated the Task 2 for 3 times. But it seems that there’s no comparison between the visual feedback ON/OFF situations. Please explain this point.

Amendments: One-way ANCOVA results shown in Table 2 include visual feedback on/off conditions. According to these results, we found significant differences for TT, NLT-fingers, and NLT-Hands between feedback on and feedback off conditions. The results are given in the Task 2 Results section and explained in the text.

R2 The benchmark condition does not simulate any hand tracking error or visual warning. So, is tracking failure allowed in actual benchmark experiments? If not allowed, how can we ensure no tracking failures happen in your experiments? In addition, what impact do tracking errors and no tracking errors in benchmark experiments have on the experimental results?

Amendments: In benchmark condition, we did not alter or change anything on the system. This includes tracking error occurrences, i.e., tracking failure allowed in benchmark conditions. This is now highlighted in the discussion section:

“We implemented the early warning method for these 3 hand tracking error conditions, and to show its effectiveness, we compared the results with a benchmark condition, where no hand tracking errors are simulated, and feedback off condition, where we simulated the errors but did not display visual feedback. In the early warning method, when a high error risk for one or more hand tracking failures was detected, we warned the users with visual feedback.”

We did not have any tracking errors since the hand tracking systems occasionally fail to detect hand movements in VR.

R2: Where does the conclusion in L650 come from? From Fig. 14 and 15? But from these two figures, it can be seen that when feedback is turned on, both Task 1 and Task 2 show a significant decrease in TT.

The conclusion in L650 comes from Figure 15 and Figure 16 (the numbers of Figure 14 and Figure 15 are now changed to Figure 15 and Figure 16, respectively). In Figure 15, we found significant differences between error types, i.e., (Low-Intensity Light – Out of Vision Hands), (Low-Intensity Light – Self Occlusion), and (Out of Vision Hands – Self Occlusion). There is no significant difference between Feedback ON and OFF on Fig. 15. On the other hand, we found a significant difference in Figure 16 (a) between Feedback ON and OFF conditions. The figure numbers are now clearly indicated in the discussion section.

R2: In L175, L177, it seems that references to figures should be enclosed in parentheses like in L189, L256, L260, etc.

Amendments: The ‘figures’ references in L175 and L177 are enclosed in parentheses.

R2: The 'conclusion' in L729 seems to be 'occlusion'.

Amendments: We thank the reviewer for their comment. We fixed this typo.

Again, we would like to thank you and the reviewers for their valuable time and comments. We hope our edits meet PlosOne’s high publication standards.

---

## [Decision Letter · Decision Letter 1]

PONE-D-25-05813R1Before hands disappear: Effect of early warning visual feedback method for hand tracking failures in Virtual RealityPLOS ONE

Dear Dr. Gemici,

Thank you for submitting your manuscript to PLOS ONE. After careful consideration, we feel that it has merit but does not fully meet PLOS ONE’s publication criteria as it currently stands. Therefore, we invite you to submit a revised version of the manuscript that addresses the points raised during the review process.

**ACADEMIC EDITOR:** Please only continue to address Reviewer 1's concerns. 

We look forward to receiving your revised manuscript.

Kind regards,

Ziyu Qi

Academic Editor

PLOS ONE

Journal Requirements:

Additional Editor Comments:

Please only continue to address Reviewer 1's concerns.

Reviewers' comments:

Reviewer's Responses to Questions

**Comments to the Author**

1. If the authors have adequately addressed your comments raised in a previous round of review and you feel that this manuscript is now acceptable for publication, you may indicate that here to bypass the “Comments to the Author” section, enter your conflict of interest statement in the “Confidential to Editor” section, and submit your "Accept" recommendation.

Reviewer #1: All comments have been addressed

Reviewer #2: All comments have been addressed

2. Is the manuscript technically sound, and do the data support the conclusions?

Reviewer #1: Yes

Reviewer #2: Yes

3. Has the statistical analysis been performed appropriately and rigorously? 

Reviewer #1: Yes

Reviewer #2: Yes

4. Have the authors made all data underlying the findings in their manuscript fully available?

Reviewer #1: Yes

Reviewer #2: Yes

5. Is the manuscript presented in an intelligible fashion and written in standard English?

Reviewer #1: Yes

Reviewer #2: Yes

6. Review Comments to the Author

Reviewer #1: The authors of the manuscript entitled "Before hands disappear: Effect of early warning visual feedback method for hand tracking failures in Virtual Reality" checked the comments of reviewers carefully and addressed the revised parts in the text completely.

The Figure 1 added to text truly but it needs modification at all because it is one of the most important figures of the article. The steps are different in context and content so the colors of the boxes must be different. The steps must be connected to each other with arrows not lines. The size of the boxes must be smaller to have a better graphical design. The texts in the boxes must be about "what the authors done in that step?". All the texts should imply an activity i.e.:

- "Implementing Statistical Data Analysis Tests" not "Statistical Data Analysis Tests".

- "Inference the recommendations and write the conclusion" not "Conclusion and Recommendations".

The other sections of the manuscript meets journal standards scientifically.

Therefore the manuscript requires minor revision again.

Reviewer #2: The authors have adequately addressed all the reviewers' comments and have additionally corrected several errors that were not previously identified. After a comprehensive evaluation of the revised manuscript in light of the publication criteria, I believe that the current version is acceptable for publication now.

7. PLOS authors have the option to publish the peer review history of their article (what does this mean?). If published, this will include your full peer review and any attached files.

Reviewer #1: **Yes: **Ali Chegini

Reviewer #2: No

---

## [Author Response · Author response to Decision Letter 2]

8 Apr 2025

Dear Coordinator,

We thank you and both reviewers for their comments. We made the following changes based on the R1 review:

The Figure 1 added to text truly but it needs modification at all because it is one of the most important figures of the article. The steps are different in context and content so the colors of the boxes must be different. The steps must be connected to each other with arrows not lines. The size of the boxes must be smaller to have a better graphical design. The texts in the boxes must be about "what the authors done in that step?". All the texts should imply an activity i.e.:

- "Implementing Statistical Data Analysis Tests" not "Statistical Data Analysis Tests".

- "Inference the recommendations and write the conclusion" not "Conclusion and Recommendations".

* We grouped the boxes and applied a color-blind-friendly palette. Now, in Figure 1, green represents the literature review, blue task implementation, orange feedback design, pink user study and data analysis, and red discussion and conclusion.

* Each step now is connected to each other with arrows.

* We re-sized the boxes to have a better graphical design.

* We edited the text to emphasize an activity, as suggested by the reviewer.

We hope that these changes meet PLOS ONE’s publication criteria.

Kind regards

Mucahit Gemici

---

## [Decision Letter · Decision Letter 2]

Before hands disappear: Effect of early warning visual feedback method for hand tracking failures in Virtual Reality

PONE-D-25-05813R2

Dear Dr. Gemici,

We’re pleased to inform you that your manuscript has been judged scientifically suitable for publication and will be formally accepted for publication once it meets all outstanding technical requirements.

Kind regards,

Ziyu Qi

Academic Editor

PLOS ONE

Additional Editor Comments (optional):

Reviewers' comments:

Reviewer's Responses to Questions

**Comments to the Author**

1. If the authors have adequately addressed your comments raised in a previous round of review and you feel that this manuscript is now acceptable for publication, you may indicate that here to bypass the “Comments to the Author” section, enter your conflict of interest statement in the “Confidential to Editor” section, and submit your "Accept" recommendation.

Reviewer #1: All comments have been addressed

2. Is the manuscript technically sound, and do the data support the conclusions?

Reviewer #1: Yes

3. Has the statistical analysis been performed appropriately and rigorously? 

Reviewer #1: Yes

4. Have the authors made all data underlying the findings in their manuscript fully available?

Reviewer #1: Yes

5. Is the manuscript presented in an intelligible fashion and written in standard English?

Reviewer #1: Yes

6. Review Comments to the Author

Reviewer #1: The Manuscript entitled "Before hands disappear: Effect of early warning visual feedback method for hand tracking failures in Virtual Reality" has been reviewed two times so far. The authors considered the reviewer's comments totally. Now the manuscript has enough merit to publish in PLOS ONE Journal.

7. PLOS authors have the option to publish the peer review history of their article (what does this mean?). If published, this will include your full peer review and any attached files.

Reviewer #1: **Yes: **Ali Chegini

---

## [Editor Report · Acceptance letter]

PONE-D-25-05813R2

PLOS ONE

Dear Dr. Gemici,

I'm pleased to inform you that your manuscript has been deemed suitable for publication in PLOS ONE. Congratulations! Your manuscript is now being handed over to our production team.

Kind regards,

on behalf of

Mr. Ziyu Qi

Academic Editor

PLOS ONE